# Perception of Need for Further Refinement in a Clear Aligner Treatment among Orthodontists, Dentists and Laypeople: A Retrospective Study

**DOI:** 10.3390/ijerph192315498

**Published:** 2022-11-23

**Authors:** Patrícia Oliveira, Iman Bugaighis, Hélder Nunes Costa, Pedro Mariano Pereira

**Affiliations:** 1Department of Orthodontics, Egas Moniz—Cooperativa de Ensino Superior CRL, Monte da Caparica, 2829-511 Almada, Portugal; 2The Libyan Authority for Scientific Research, Tripoli P.O. Box 80045, Libya; 3Centro de Investigação Interdisciplinar Egas Moniz, Egas Moniz—Cooperativa de Ensino Superior CRL, Monte da Caparica, 2829-511 Almada, Portugal

**Keywords:** additional aligners, refinement, dentists, orthodontics, perception

## Abstract

Clear aligner treatment often requires further refinement to improve the orthodontic treatment outcome. However, the perceptions of treatment outcomes evaluated by orthodontists and dentists are sparse, and laypeople’s perceptions have yet to be explored. Here, we explore the perceptions of orthodontists, dentists, and laypeople concerning the treatment outcomes achieved after completing the first sequence of aligners. This cross-sectional study involved 37 orthodontists, 67 dentists, and 93 laypeople. We administered an online questionnaire containing intra-oral photographs of nine completed cases with pre- and post-first sequences of aligners. As a control, we used a digital prediction system for the treatment outcome. Self-perception was reported using a visual analog scale. Both orthodontists and dentists had similar perceptions about treatment outcomes (*p* = 0.363) but significantly differed from laypeople (*p* ≤ 0.0001). Both orthodontists and dentists recommended further treatment; orthodontists were more critical than dentists (*p* ≤ 0.001). Orthodontists were more critical than dentists in their evaluations of the need for further treatments; however, their perceptions of treatment outcomes were similar. Laypeople were more satisfied with the treatment outcomes, were less concerned with occlusion, and were more focused on the aesthetic results of the treatment.

## 1. Introduction

The concept of clear aligners was proposed in 1940 when Kesling [1] introduced the positioner to refine the final stages of orthodontic treatment. In 1998, Align Technology (Santa Clara, CA, USA) introduced Invisalign, a series of removable polyurethane aligners, a transparent, aesthetically pleasing alternative treatment modality to conventional fixed appliances for mild to moderate malocclusion cases. The introduction of auxiliary tools integrated with the clear aligner treatment (CAT) has improved treatment outcome capacity, predictability, and stability. Several investigators reported that the clear aligner system can effectively move teeth (simultaneously) and treat a variety of malocclusions [2,3,4]. Evolving computer graphics technology and growing research advancements have led to the integration of digital diagnostic tools, programmed virtual treatment plans, and biomechanical designs using ClinCheck^®^ Pro 5.7 software. This software offers a three-dimensional (3D) visual interface, allowing clinicians to customize treatment plans, anticipate and monitor treatment progress, and implement modifications when required [5,6]. Frequently, aesthetic and functional treatment objectives with clear aligners are not attained after the first set of aligners. Therefore, it is necessary to additionally refine the sequence of aligners [2].

Although the professional use of clear aligners continues to increase, doubts regarding their effectiveness and efficiency remain [7]. A recent systematic review of 20 studies comprising randomized and non-randomized clinical trials, as well as cohort and case-control studies, concluded that treating complex malocclusions with clear aligners is possible, but the results are less accurate than those achieved with conventional fixed appliances [8].

Finishing is a critical part of orthodontic treatment. Tooth morphology variations, such as round teeth, require additional CAT to achieve the desired alignment. Tooth movement also depends on the response of the periodontal tissues to an applied force [9]. Biological response variations to tooth movements can affect treatment outcomes and the need for further refinement [10].

Over the years, a significant body of related research has exclusively focused on the effectiveness of tooth movements with a CAT [11,12,13,14]. The reported discrepancy between predicted and achieved results is believed to be around 50%, requiring several stages of refinement or additional treatments [4]. Many orthodontists have reported that 70% to 80% of their patients require mid-treatment reevaluations that might necessitate additional sequences of aligners or continuation of the treatment with fixed appliances [15]. However, the perceptions of treatment outcomes evaluated by orthodontists and dentists are sparse and the perceptions of laypeople regarding aligner treatment outcomes have yet to be explored [2,4,16,17].

A recent survey found several differences in the treatment planning, training, and knowledge between orthodontists and dentists performing CAT [18]. Although, many dentists have enrolled in short-term continuing education courses aimed at mastering the concept of treatments with clear aligners, their competencies in diagnosing and adequately treating patients remain controversial [19,20].

Annually, the clear aligner market expands and more potential orthodontic patients worldwide prefer to be treated with this technique compared to conventional appliances. This can be attributed to aesthetic reasons, ease of use, simpler oral hygiene maintenance, and comfort of wear [21]. Chhibber et al. [22], in their prospective randomized control trial, noted similar oral hygiene statuses among conventional, self-ligated brackets and clear aligner therapies. Furthermore, Sfondrini et al. observed no significant differences between the microbiota compositions in the oral cavities of patients treated with aligners and controls for the first two months of treatment [23]. However, most individuals seeking CAT only take into account the aesthetic component of the smile (especially in the anterior region). However, for both the orthodontist and the patient, it is important to finish a case with good esthetics and acceptable functional occlusion [24,25].

Detailed treatment prediction is a challenge for orthodontists and many aligner companies [4]. Accurate digital case planning is one of the key elements to a successful treatment, and it is essential to avoid the need for additional aligners. However, very little is known about whether the need for additional refinement is shared by patients and other dental professionals [26,27].

The aim of this study was to explore the perceptions of orthodontists, dentists, and laypeople concerning the treatment outcomes achieved after completing the first sequence of aligners.

## 2. Materials and Methods

This was a cross-sectional, retrospective, questionnaire-based survey carried out at the Orthodontic Department, Egas Moniz University Institute, Portugal. Ethical approval was granted and the questionnaire was approved by the ethics committee of the university (nº 1008). Informed consent was obtained from participants whose intra-oral photographs were used in this study.

A web-based survey questionnaire was developed to evaluate the perceived need for further treatment with Invisalign following the first sequence of aligners. The online survey tool Google Forms was used to establish the questionnaires. A thorough process of the questionnaire preparation was conducted. During the question formulation, we considered inviting laypeople with no prior knowledge of orthodontic treatment to be among the raters. We aimed to develop simple, focused questions directly related to the study’s objective. During the development process, the questionnaire was rated by two orthodontists, two dentists, and two laypeople. The feedback from the orthodontists and dentists was positive. However, the questions needed to be more straightforward for the laypeople, who felt that they needed more information to base their ratings. The updated questionnaire, guided by logical feedback from the participants, was re-rated. At this stage, the laypeople were more confident about their evaluations, and the other groups of raters found that the updated questionnaire remained within the scope of the study objective. All participants in the questionnaire development were excluded from the following rating process.

The first section of the questionnaire consisted of sociodemographic information, including profession, sex, and age. The second section included a set of nine treated clinical cases to assess the respondent’s perceived need for additional treatment. The duration of the treatment ranged between 6 and 8 months. Only clear aligners with attachments were used for the cases without auxiliary mechanics or interproximal stripping. Each case was presented with three pretreatment intra-oral photographs (Figure 1, top) and three intra-oral photographs of the post-first sequence of aligner treatments (Figure 1, bottom). The middle of Figure 1 displays the simulated treatment outcome planned by the ClinCheck**^®^** software (control). All photographs were taken by the same operator using a Canon EOS 550D camera, with a macro 100 mm canon lens, and a flash macro ring lite MR-14EX II following the same protocol: frontal, right, and left centric occlusion views. All cases were treated by the same clinician between 2018 and 2020 with the Invisalign system. The nine cases represent different types of dental malocclusion: three cases with sagittal discrepancies (one case with half-unit pre-molar class II and two cases with half-unit pre-molar Class III), three cases with vertical discrepancies (one case with a deep overbite of more than two-thirds of the lower incisor crown and two cases with anterior open bites of 1 mm). The last three cases consisted of transversal discrepancies without crossbites associated with crowding (4.1 to 8 mm of crowding).

The cases were randomly redistributed to establish three different questionnaires regarding the presented case sequences but were similar in content.

For each case, there was a set of three questions: “How do you rate the result achieved?”, “Do you think additional treatment is necessary?” and “Do you think the expected result was achieved?”. Each rater was asked to mark his answer on a visual analog scale (VAS) located under each photograph. The VAS scores were between 0 and 10, with 0 corresponding to “Poor” and 10 to “Excellent” for the first question, “Not necessary” and “Very necessary” for the second question, and “Not at all achieved” and “Totally achieved” for the third question.

An assessment of the orthodontic treatment outcome with a quantitative index might help one to establish goals, evaluate effectiveness, and achieve a measurable finish for orthodontically treated cases [28]. Several quantitative indices have been developed to evaluate malocclusion severity, orthodontic treatment need, and treatment outcome [29,30,31,32]. The peer assessment rating (PAR) index has been widely used to provide a single summary score for occlusal discrepancies, which may be found in malocclusions [32,33,34,35,36].

In this study, the PAR index was used to assess the pretreatment needs and relate them to the post-first phase treatment of a CAT. Table 1 shows the pre- and post-treatment PAR values for all of the included cases as well as the reduction percentages of the PAR values. It can be seen that the PAR values decreased for all of the cases. The 3D virtual models were used for the PAR assessment. A reevaluation of the PAR index was conducted after a two-week interval to assess the consistency and reproducibility of intra-operator occlusal trait measurements. Table 1 shows that the PAR index was reduced for all cases. The intraclass correlation coefficient (ICC) was found to be greater than 0.90, indicating an excellent level of reproducibility between both trials. Moreover, the paired t test revealed no statistically significant differences between the two trials (*p* ≥ 0.298).

Raters consisted of three groups: orthodontists, dentists, and laypeople. All 80 registered specialists in the Portuguese Orthodontics Association were invited to participate in the study. A total of 120 questionnaires were sent to dentists and 120 were delivered to laypeople. The dentists were selected randomly from a list of members of the multidisciplinary Egas Moniz university clinics and laypeople consisted of non-orthodontic patients attending the multidisciplinary Egas Moniz university clinics. The invitations were sent by email and the responses were received between January and February 2022. A reminder email was sent to the orthodontists two weeks later, requesting their participation in case they had not yet completed the questionnaire.

IBM Statistical Package for Social Science (SPSS) v.27 software was used for the present analysis. For all linear variables, the Shapiro–Wilk test revealed that the data were significantly different from the normal distribution. Levene’s test established the variances as non-homogeneous. Nonparametric tests were applied to compare the participants’ responses between the three groups. The Spearman’s rank correlation coefficient test was employed to evaluate the raters’ responses’ with regard to age and sex. Multiple comparisons between the variables and the raters in the three groups were conducted using Kruskal–Wallis and Mann–Whitney analyses. In the Mann–Whitney test, the Bonferroni correction was applied to control the type I error rate. This was conducted by dividing the critical *p*-value for significance (*p* < 0.05) by the number of groups included in the study (three). Therefore, the level of significance was adjusted to 0.016.

## 3. Results

Out of the 80 orthodontists invited to participate in the study, 37 participated. A total of 67 dentists filled out and returned the questionnaire. The laypeople consisted of 93 non-orthodontic patients. Table 2 displays the descriptive data of the participants in the three groups. Almost half of the participants in the dentist and laypeople groups were under the age of 30 (51% and 49%, respectively). A total of 62% of the orthodontists were between 41 and 60 years old. Furthermore, most of the raters in the three groups were females (orthodontists: 57%; dentists: 75%; laypeople: 71%).

Table 3 presents the average median and the interquartile (25% and 75%) scores of the evaluators for the three different questions. Kruskal–Wallis analyses revealed that there were statistically significant differences between the three groups (*p* < 0.0001).

The spearman’s rank correlation coefficient revealed a moderately negative correlation between the raters´ responses, age (r = −0.4, *p* = 0.033), and sex (r = −0.3, *p* < 0.001), indicating that older raters and females were more critical compared to younger and male participants. The Mann–Whitney pairwise analysis revealed a statistically significant difference in the evaluation of the need for further treatment between the three groups. Both orthodontists and dentists had similar insights into treatment outcomes at *p* = 0.363 (Table 4), which significantly differed from the perceptions of laypeople (*p* ≤ 0.0001). Furthermore, there were statically significant differences among the three groups regarding the response to the second question (*p* ≤ 0.001). Although both the orthodontists (median = 9) and dentists (median = 8) recommended further treatment, the orthodontists were significantly more critical compared to the dentists (*p* ≤ 0.001). On the contrary, laypeople were less concerned with continuing the treatment (*p* ≤ 0.001). Regarding the third question, the orthodontists and dentists observed significant discrepancies between the treatment outcome of the first phase of aligners and the simulation provided by ClinCheck**^®^** Pro 5.7 (*p* ≤ 0.001). On the other hand, for the laypeople, the treatment outcome was more similar to the corresponding treatment outcome simulations compared to the other groups (*p* = 0.370).

## 4. Discussion

This was a retrospective questionnaire-based investigation that compared the subjective perceptions among three groups of raters (orthodontists, dentists, and laypeople) concerning the need for additional refinement after the first sequence of a CAT. Recently, there has been a significant amount of literature evaluating the quantitative outcomes of a CAT. However, qualitative research in this area is sparse. To our knowledge, this study was the first investigation that compared the perceived need for additional treatment in a CAT.

There were discrepancies in the number and sex of the recruited raters in each group. This was due to the high non-respondent rate among the approached orthodontists, despite a two-week follow-up email requesting their participation in case they still needed time to fill out the questionnaire. The Spearman’s correlation coefficient revealed that older raters and females were more critical to the CAT outcome compared to younger and male participants. However, due to the present study’s limitations, this outcome has to be interpreted with caution. Researchers could take advantage of orthodontic conferences by informing orthodontists about the investigators’ research and the importance of participating to boost the response rate and even out male/female distribution.

The limitations of treating complex malocclusions with a CAT are described in the literature, for which there is an incompatibility between the predicted treatment outcome suggested by the ClinCheck**^®^** software and the achieved results [37]. Tooth movement with a CAT can be more complex due to the absence of specific points of force application, tooth anatomy, properties of the aligner material, sliding movements between contact points, and other biomechanical factors [11,38]. The reported aspects lead to the tipping of the clinical tooth crown with lesser root movement. According to Zhang et al. [39], the average discrepancy observed between the achieved crown and root movement in the maxilla is approximately 2.062 mm and in the mandible is 1.941 mm.

In our study, both orthodontists and dentists agreed that the treatment outcome was not satisfactory enough (*p* = 0.363), contradicting the perception of laypeople who were satisfied with the results. The three groups of raters significantly differed in evaluating the need for further refinement (*p* ≤ 0.0001). The laypeople were satisfied with the achieved treatment, while the dentists were significantly less satisfied with the outcome than the laypeople. The orthodontists were the most critical of the three groups. This might be due to their thorough knowledge of occlusions and the significance of achieving a stable inter-arch relationship to limit post-treatment relapse.

In this investigation, both orthodontists and dentists agreed that significant differences existed between the achieved treatment outcomes and the simulation predicted by the ClinCheck**^®^** software. Contrarily, laypeople were less capable of differentiating between the treatment simulation by the ClinCheck**^®^** software and the clinical results obtained. Our findings agree with the work by Heath et al. [26], who compared the perceptions of orthodontic case complexities among orthodontists, orthodontic residents, dental students, and dentists with the American Board of Orthodontics Discrepancy Index (DI). They concluded that formal orthodontic training and exposure to the specialty impacted the participants’ abilities to identify case complexities in moderate to severe cases.

The PAR index is an occlusal assessment index used for measuring the deviation from normal occlusions and quantitatively evaluating orthodontic treatment standards by comparing pre-treatment and post-treatment dental casts [33,34,35]. In this investigation, we applied the PAR index to assess the pre-treatment need and relate it to the post-first phase treatment outcome. All cases started with low to moderate PAR index values (3 to 9). Following treatment with the first sequence of aligners, the PAR index values of all cases improved by more than 70%. This improvement is considered an acceptable outcome according to the related literature [36]. Curiously, the three evaluator groups had similar favorable ratings for the cases with crowding and with less need for orthodontic treatment as revealed by the PAR index. All three groups agreed that alignment and occlusion treatment outcomes were acceptable, with no need for additional aligners, assuming that what was simulated in ClinCheck**^®^** was achieved. We believe those discrepancies were so minimal in the pre-treatment stage and improved significantly after the first phase of treatment with a CAT. Future studies presenting an increased number of treated cases might explore the correlation between the PAR score change and grader satisfaction with the treatment outcome.

Finishing a case with acceptable intra-arch and inter-arch relationships is critical for post-treatment stability. Kuncio et al. [40] conducted a comparative retrospective cohort study evaluating post-orthodontic treatment relapse between a group of patients treated with conventional fixed appliances with a matched group treated with Invisalign. They analyzed the dental casts and panoramic radiographs using the Objective Grading System (OGS) of the American Board of Orthodontics at the deboning stage and three years post-retention. The authors concluded that the Invisalign group had a greater anterior and total mandibular alignment relapse post-retention compared to those treated with fixed appliances. Their study interpreted the greater relapse in the Invisalign group because the 2-week interval of the aligner system was too short and might have led to poor bone formation and more relapse. Katsaros et al. reported the importance of a proper interproximal contact area to provide stability after the orthodontic treatment. In our study, incisal tooth morphology was preserved, and none of the patients had interproximal stripping. A further study exploring the relapse extent in cases treated with and without interproximal stripping would be informative. However, when the CAT outcome is unsatisfactory, diastema management with esthetic restorative materials could help post-orthodontic stability and provide pleasant esthetic outcomes [41,42,43].

Based on our results, laypeople’s concerns were aesthetically driven; more precisely, they focused on the alignment of the anterior teeth. Several studies evaluating the perceptions of laypeople and professionals on altered smile attractiveness reached similar conclusions. Laypeople can identify various factors affecting smile esthetics but are less critical than orthodontists and dentists when it comes to posterior inter-arch and intra-arch relationships [44,45,46,47,48]. Therefore, it would be interesting for a future study to include both extra-oral and intra-oral photographs of more cases with a wide variety of malocclusions treated by different orthodontists to compare the perceptions among the three groups of raters.

The consistent promotion of the CAT, driven by stakeholder companies, and increased public familiarity with more aesthetic treatment options have expanded their use [49]. Mostly, the CAT is a patient-driven treatment and it is here to stay. However, orthodontists need to make the patients aware of the scope and limitations of a CAT, supported by evidence. Moreover, orthodontic treatments have to be continued until stable functional occlusions are achieved even when the patients are satisfied with the esthetics. Consenting patients who do not want to continue with their treatments are advised to avoid the responsibility of possible relapses.

There are certain limitations in this retrospective analysis. Firstly, a possible mood bias might have influenced the raters’ evaluations. Moreover, the research consisted of a relatively small sample size with unequal numbers of male/female participants. Further robust studies could expand the present investigation and use our results for power calculations. Furthermore, the inclusion criteria for the included cases could be more specific, e.g., by including other malocclusion types. Moreover, there were significant differences in the ages between the groups, with orthodontists being the oldest. This could reflect the greater knowledge and experience in orthodontics and the evaluation of treatment outcomes.

## 5. Conclusions

Orthodontists were more critical than dentists in evaluating the need for further treatment; however, their perceptions of the treatment outcomes were similar. Laypeople were more satisfied with the treatment outcomes, less concerned with occlusion, and were more focused on the aesthetic results of the treatment.

## Figures and Tables

**Figure 1 ijerph-19-15498-f001:**
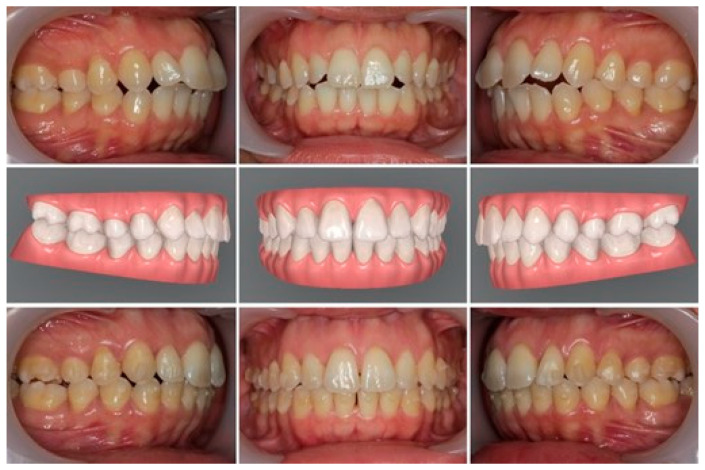
Representation of one of the cases included in the questionnaire; (**top**) pretreatment intraoral photographs; (**middle**) digital simulation (ClinCheck^®^) at the end of the first sequence of the aligners; (**bottom**) intraoral photographs after the first sequence of aligners.

**Table 1 ijerph-19-15498-t001:** Pretreatment peer assessment rating (PAR) index after the first sequence of aligners, the PAR index for each case, and the percentage of the PAR index reduction.

Cases	Initial PAR	After 1st Sequence of Aligners, PAR	Percentage of PAR Index Reduction (%)
1	7	1	85.7%
2	8	2	75%
3	6	1	83.3%
4	3	1	66.6%
5	8	2	75%
6	3	1	66.6%
7	9	1	88.8%
8	4	1	75%
9	5	0	100%

**Table 2 ijerph-19-15498-t002:** Demographic distributions of the three groups of raters (orthodontists, dentists, laypeople).

Variables		Orthodontists	Dentists	Laypeople
Number		37	67	93
Age	<30	1	34	46
	30–40	9	23	11
	41–50	12	9	10
	51–60	11	1	16
	>61	4	0	10
Gender	Female	21	50	66
	Male	16	17	27

**Table 3 ijerph-19-15498-t003:** Median scores and interquartile value of each group of evaluators for the three questions.

	Orthodontists		Dentists		Laypeople	
Questions	Median (50%)	IQR (25–75%)	Median (50%)	IQR (25–75%)	Median (50%)	IQR (25–75%)
1st	6	5–8	7	5–8	8	7–9
2nd	9	7–10	8	6–10	7	4–8
3rd	6	4–8	6	4–8	8	6–9

**Table 4 ijerph-19-15498-t004:** Comparisons of the perceptions among the different groups of evaluators for the three different questions at *p* < 0.016.

	Orthodontists vs. Dentists	Orthodontists vs. Laypeople	Dentists vs. Laypeople
Questions	*p* value	*p* value	*p* value
1st	0.363	0.0001	0.0001
2nd	0.0001	0.0001	0.0001
3rd	0.370	0.0001	0.0001

## Data Availability

Data will be available upon request.

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
