# Peer review of "Perception of Need for Further Refinement in a Clear Aligner Treatment among Orthodontists, Dentists and Laypeople: A Retrospective Study"

_ijerph, 2022, doi:10.3390/ijerph192315498_

Round 1

Reviewer 1 Report

Orthodontic treatment is an important element of dental treatment. When planning orthodontic treatment, important indications for its conduct should be taken into account, i.e. skeletal and dental defects and improved aesthetics. Orthodontic treatment may be performed with the use of removable orthodontic appliances and fixed appliances. A new alternative is the use of orthodontic aligners. The aesthetic aspect of orthodontic treatment largely depends on the awareness of the doctor and the patient. The authors of the work raised a very important topic, which is the perception of need for further refinement in a clear aligner

treatment among orthodontists, dentists and laypeople. From the point of view of a practicing oral surgeon, there are real feelings about the treatment needs and therapeutic success according to the treatment effect. The authors of the work unequivocally show that both orthodontists and dentists agreed that the treatment satisfied outcome was not satisfactory enough, contradicting the perception of laypeople who were with the results. The conclusion of the work is a conclusion that the orthodontists were more critical than dentists in their evaluation of the need of further treatment, however their perception of the treatment outcome was similar. While, laypeople were more satisfied with the treatment outcome and were less concerned with occlusion and more focused on the aesthetic result of the treatment. It is also worth mentioning that would seem that laypeople concerns were aesthetically driven, more precisely, their focus went on the alignment of the anterior teeth. The authors used a web-based survey questionnaire was developed to evaluate the perceived need of further treatment with Invisalign following the first sequence of aligners. The online surveying software Google Forms was used to stablish the questionnaires. In my opinion, this is an additional advantage of the conducted research, because this form can be used to conduct multicentre research and draw conclusions based on the analysis of large groups of respondents. The work was written in correct English. The research methods used are adequate and correct and I believe that the work fits perfectly with the subject of the works published in the International Journal of Environmental Research and Public Health International Journal of Environmental Research and Public Health.

Author Response

Dear Reviewer,

Highly appreciated feedback. Thank you very much for your valuable positive review.

Yours sincerely,

The Authors

Reviewer 2 Report

Manuscript of considerable interest for the orthodontic sector, as the use of invisible aligners is increasingly in use for

corrections of maloclusions.

Before proceeding with the evaluation it is essential to carry out a major revision.

Abstract well described, emphasize the results even more

keywords, enter them with specifications

Introduction

insert as already published by Prof. Sfondrini et al that there is no difference in the microbiota in patients with aligners rather than without, and the cleansing systems studied by Prof. scribante et al.

Materials and methods, enter the sample interest

results: highlight significant data

Discussion, focus on evaluation together with patients to make lay people understand the evolution of therapy

Bibliography adds required references

Author Response

Dear Reviewer,

We are very grateful for your comments and those of your reviewers. We have attempted to address these in detail below and agree that they have assisted in improving our work. All responses and adjustments in the article are tracked in the main document.

Yours sincerely,

The Authors

Reviewer 3 Report

Dear Authors,

generally, the write-up of the paper is good. Here are some suggestions in order to improve the manuscript.

Here are some suggestions:

1- 

Research hypotheses shall be declared at the end of the Introduction, they should be labeled as such and provide a numerical listing of each hypothesis (even just one). This listing is key to the paper. The same sequence of hypothesis testing will be used to structure the Materials and Methods, Results, Discussion, and Conclusions sections. Hypothesis shall be accepted (or rejected) in the Discussion.

2-

As correctly stated at the end of the discussion, one of the limitations of this study is related to the low number of participants, especially orthodontists. Another limitation that could be outlined is that the female/male ratio in Dentists and Laypeople is strongly shifted through females.

3-

Line 90: add the State/Country of the University

4-

Lines 101-3:

The ClinCheck images should be the second not the third (if Figure 1 is correct). Please adjust either the text or the Figure order and the legend.

5-

Discussion

This section should be expanded. Discussion is generally longer and covers more arguments related to the topic and the results of this research.

6-

Line 275:

relapse shall be also related to lack of interproximal stable contacts, which generally occurs with CAT.

The authors could add the following sentence to support this important concept:

7-

Katsaros et al. reported the importance of a proper interproximal contact area to provide stability after the orthodontic treatment (suggested reference: PMID: 23197574). When CAT outcome is unsatisfactory, diastema management with esthetic restorative materials could help post-orthodontic stability while providing esthetic pleasant outcomes. (suggested references: part1 and 2 of same article: PMID: 25223143 and PMID: 25975063)

The above suggested sentence and references could be also placed between line 240 and 241. The reviewer leave to the authors the decision on where to place it.

Table 2:

There is an error in the Dentists column. If the numbers are added together the value exceed 67 (34+23+9+1+34). 

Author Response

(The authors gave the same response as above.)

Round 2

Reviewer 2 Report

The manuscript has been correctly revised, it can be published

Author Response

(The authors gave the same response as above.)
